# Intranasal Borna Disease Virus (BoDV-1) Infection: Insights into Initial Steps and Potential Contagiosity

**DOI:** 10.3390/ijms20061318

**Published:** 2019-03-15

**Authors:** Alexandra Kupke, Sabrina Becker, Konstantin Wewetzer, Barbara Ahlemeyer, Markus Eickmann, Christiane Herden

**Affiliations:** 1Institute of Veterinary Pathology, Justus Liebig University, 35392 Giessen, Germany; sabrina.becker@vetmed.uni-giessen.de; 2Institute of Virology, Philipps University, 35043 Marburg, Germany; eickmann@staff.uni-marburg.de; 3Institute of Functional and Applied Anatomy, Hannover Medical School, 30625 Hannover, Germany; Wewetzer.Konstantin@mh-hannover.de; 4Institute for Anatomy and Cell Biology, Division of Medical Cell Biology, Justus Liebig University, 35385 Giessen, Germany; Barbara.Ahlemeyer@anatomie.med.uni-giessen.de

**Keywords:** borna disease virus, initial phase, olfactory epithelium, olfactory ensheathing cells, OECs, in vivo, in vitro

## Abstract

Mammalian Bornavirus (BoDV-1) typically causes a fatal neurologic disorder in horses and sheep, and was recently shown to cause fatal encephalitis in humans with and without transplant reception. It has been suggested that BoDV-1 enters the central nervous system (CNS) via the olfactory pathway. However, (I) susceptible cell types that replicate the virus for successful spread, and (II) the role of olfactory ensheathing cells (OECs), remained unclear. To address this, we studied the intranasal infection of adult rats with BoDV-1 in vivo and in vitro, using olfactory mucosal (OM) cell cultures and the cultures of purified OECs. Strikingly, in vitro and in vivo, viral antigen and mRNA were present from four days post infection (dpi) onwards in the olfactory receptor neurons (ORNs), but also in all other cell types of the OM, and constantly in the OECs. In contrast, in vivo, BoDV-1 genomic RNA was only detectable in adult and juvenile ORNs, nerve fibers, and in OECs from 7 dpi on. In vitro, the rate of infection of OECs was significantly higher than that of the OM cells, pointing to a crucial role of OECs for infection via the olfactory pathway. Thus, this study provides important insights into the transmission of neurotropic viral infections with a zoonotic potential.

## 1. Introduction

Mammalian Bornavirus 1 (BoDV-1) is the prototype of the family *Bornaviridae*, and contains a single-stranded, linear, and negatively orientated genome [1,2,3]. The main natural hosts, horses and sheep, develop a non-purulent meningoencephalitis [4,5,6], and infection leads to viral persistence [7,8]. Clinical signs include abnormal behavior; sensory alterations or loss of motor functions; and, during late stages, lethargy, somnolence, and stupor, as well as ataxia and paresis, comparable to the recently identified human cases. Death usually occurs within four weeks after the onset of clinical signs [5].

In addition to classical mammalian BoDV-1, many new genetically distant orthobornaviruses have been detected in different species, ranging from psittacine birds, water fowl, reptiles, and squirrels [9,10,11,12]. A novel Bornavirus detected in variegated squirrels (VSBV-1) differs significantly from BoDV-1 on a genomic level, and is responsible for four fatal human encephalitis cases [11,13]. Recently, three cases of BoDV-1-infected transplant recipients, of whom two died, have underscored its potential role as a zoonotic human pathogen [14]. Many neurotropic viruses belonging to the order *Mononegavirales* use the olfactory pathway to enter the central nervous system (CNS) [15,16,17,18,19]. This pathway is extraordinary, because the olfactory epithelium is the only site of the body where neurons are in direct contact with the environment, and a timely and defensive immune response seems to be lacking [19]. The intranasal infection represents an assumed major route of entry for BoDV-1. In rat models, the spread of BoDV-1 to the CNS has already been demonstrated after intranasal infection [15,20]. Other routes, for example, subcutaneous infection, have been reported but are less efficient [21]. Interestingly, BoDV-1 employs unique strategies such as nuclear transcription and replication in order to establish a neurotropic, non-cytolytic, and persistent infection [5,22]. However, data on the necessity of the initial transcription and replication at the entry site and at susceptible cell types have so far been addressed only for intracerebral infection, where neurons seem to provide the most efficient replication site [23], but not for the intranasal route.

In previous studies, the intranasal infection of immunocompetent rats with BoDV-1 at the age of four or five weeks resulted in clinical signs such as a lack of coordination, apathy, reduced food intake, and emaciation, starting between 18 and 24 days post infection (dpi) [15,21]. The case fatality rate reached about 90% within one week after the onset of clinical signs [15]. Histopathologically, the animals developed inflammatory and edematous changes in the brain, but not in the olfactory epithelium. In contrast to the intracerebral infection, areas of necrosis and edema were found in the grey matter, as well as infiltrates composed mainly of macrophages. As a consequence, small cyst-like structures in a variety of CNS areas belonging to the olfactory system have been noted [15].

The intranasal infection of the immunocompetent rats most likely reflects the situation in end- or accidental-hosts, such as horses, sheep, and even humans. Here, infection runs a strict neurotropic course. In contrast, the infection of reservoir species, such bicolored white tooth shrews and possibly variegated squirrels, leads to a disseminated virus distribution without inflammatory lesions or clinical signs [24,25]. Which route of transmission plays the most important role in these animals needs to be addressed, and the presence of the virus in the nose as well as in many secretions, excretions, and skin scales, could point also to the role of intranasal transmission [25,26].

To date, the role of the olfactory ensheathing cells (OECs) for the transmission of viruses to the CNS remains unknown. These cells guide the olfactory nerve fibers along their way to the CNS, and fulfill glia-like functions [27]. They are most often used to study the regeneration of, for example, spinal cord injuries [28], and their role for viral propagation has so far only been addressed for the human herpesvirus-6 [29]. Either direct infection or the formation of channels for the transmission of viruses to the CNS has been discussed [17,30]. In this study, we compared the intranasal infection of Lewis rats with a primary culture of the rat olfactory epithelium in order to get insight into the initial phase of the infection, with BoDV-1 as a model for neurotropic infections that enter the CNS via the olfactory route. Even after decades of research on BoDV-1, it is still unclear whether an initial replication and transcription takes place in the olfactory mucosal (OM) or not. In order to address the role of the OM and OECs during intranasal infection with BoDV-1, we compared their susceptibility to BoDV-1, in order to evaluate their impact on the initial steps of infection.

## 2. Results

### 2.1. Clinical Findings and Histopathology

Between 3 h post infection (hpi) and 21 dpi after the intranasal infection of immunocompetent Lewis rats, no neurological disorders or any other clinical signs were observed.

The nasal tissue and complete brains of all of the infected and Mock-infected rats were examined for every timepoint. No neurodegenerative lesions were detectable. However, a mild non-suppurative meningitis was present at 21 dpi in the infected rats.

### 2.2. Detection of BoDV-1-N, BoDV-N-1-Specific Genomic RNA (gRNA), and mRNA

The presence of Mammalian Bornavirus 1 nucleoprotein (BoDV-1-N) was determined by immunohistochemistry with an antibody directed against the nucleoprotein (Bo18), and was characterized by a granular to diffuse nuclear, or a patchy to diffuse cytoplasmatic staining (Figure 1). This pattern was already found in a few areas of the olfactory epithelium at four dpi (Figure 1A), and was seen multifocally distributed at later stages (Figure 1F). In the nerve fibers of the lamina propria, patchy or streaky staining was detected. A viral antigen was detected for the first time four dpi in two of the five rats in the single adult olfactory receptor neurons (ORNs). Moreover, in one rat, it was also seen in the single juvenile ORNs. At seven dpi, BoDV-1-N was visible for the first time in the single globose basal cells in one rat (Figure 1B). From 14 dpi onward, in all of the rats, extensive staining for BoDV-1-N was present in the adult ORNs. Additionally, in all of the rats, BoDV-1-N was found in juvenile ORNs, even though this was in fewer cells. In three of the five animals, BoDV-1-N was further detected in the sustentacular cells and globose basal cells, as well as in the horizontal basal cells in tow of the five rats. Furthermore, in all of the rats, a distinct staining was observed in the nerve fibers of the lamina propria and in the OECs, either in the single cells or nerve fibers, but in most cases, in more than 60 OECs or nerve fibers (Figure 1C–G; Appendix A). At 21 dpi, a strong staining was visible in all types of cells of the OM, and also in the nerve fibers and OECs in all of the rats (Figure 1F,G). In the brain, BoDV-1-N was first detectable at 14 dpi, mainly in the olfactory bulb. However, the single pyramidal cells of the hippocampus were also infected. Already at 21 dpi, the infection had spread to all of the regions of the brain.

In addition to immunohistochemistry, in situ hybridization was used to examine whether viral replication and transcription takes place in the olfactory mucosa, or if it is only used as an entry site and spreads to the brain without significant multiplication. Signals indicating the presence of the BoDV-1-N viral genomic or mRNA resembled the distribution of viral antigens. Signals for gRNA comprised single or several dark blue to black granules only in the nucleus (Figure 2). This staining was more intense at later time points, resulting in a nearly diffuse nuclear staining in the positive cells. BoDV-1 gRNA was detected first at seven dpi in three of the five rats in mature ORNs in low or in moderate numbers (Figure 2A,E). At this time, gRNA was also seen in two of the rats in some juvenile ORNs (score of 1 and 2) and in single nerve fibers. In one rat, gRNA could also be detected in OECs (Figure 2B). At 14 dpi, gRNA was detected in all of the five rats in mature and immature ORNs and OECs. The number of positive OECs ranged between a score of 1 and 2 in all of the rats. In two rats, gRNA was also present in the single nerve fibers of the lamina propria (Figure 2C). After 21 dpi, the number of positive ORNs increased in mature and juvenile ORNs, as well as in the nerve fibers and OECs (Figure 2D,E; Appendix A).

The signals for viral mRNA resulted in a granular, but predominantly strong and diffuse staining of the cytoplasm of the infected cells (Figure 3). Occasionally, an additional granular staining of the nucleus was seen, especially at early time points. BoDV-1 mRNA was first detected at four dpi in two of the five rats in single adult ORNs in the cytoplasm (Figure 3A,E). At seven dpi, the number of signals in the ORNs increased to a score of 1 or 2 in four of the five rats (Figure 3E). At 14 dpi, mRNA was present in mature and immature ORNs, mostly in a multifocal distribution pattern, in all of the rats. At this time, mRNA was also found in the nerve fibers and OECs in two of the five rats (Figure 3B,E). At 21 dpi, the number of positive mature ORNs, nerve fibers, and OECs further increased, whereas the number of juvenile ORNs remained at a high number (Figure 3C–E; Appendix A).

As for BoDV-1-N, first, only few areas of the olfactory epithelium contained positive cells for BoDV-1-N gRNA and mRNA. Until 21 dpi, the number of infected foci increased steadily, leading to a multifocal distribution pattern (Figure 2E and Figure 3E). By an exact Kruskal–Wallis test, it was confirmed that the presence of BoDV-1-N, gRNA, and mRNA for all cell types and structures was dependent on the time of post infection (*p* < 0.05), indicating an increase of viral replication and transcription until 21 dpi (Appendix A). Moreover, this increase followed a monotonous trend as determined by Spearman’s rank correlation, corresponding to a strong interaction between the time post infection and the number of positive cells or structures, respectively (Appendix A).

### 2.3. Characterization and BoDV-1 Infection of the Culture of the OM

In the primary culture prepared from the rat olfactory mucosa, the typical bipolar ORNs were identified by a positive β III-tubulin immunoreactivity. They were found either singularly or in small groups (Figure 4A). The dendritic knob was often clearly visible. Another conspicuous aspect was that the axon and dendrite did not stretch out to full extend before one day in culture. During earlier observations, these processes lay close to the cellular body.

The percentage of ORNs was 33% 4 h after seeding, and decreased to 27% after 24 h (Figure 4B). A further decline was observed until at seven dip, where only 6% neurons remained.

Infected cells were recognized by the presence of the BoDV-1-N protein in distinct intranuclear granules of varying sizes, which were not present in the negative control, or by a strong and diffuse staining of the cytoplasm (Figure 5A). At four dpi, a mean of 11% of the cells were infected, with an increase in the following investigations points (Figure 5B). At seven dpi, 19% of the cells were infected, while this was the case for 23% at 10 dpi, and about 29% at 14 dpi. This indicated a steady spread of the infection in the culture system. Strikingly, while at the early timepoints mostly single cells were infected, the infected cells clustered together in nests at later timepoints, hinting at a cell-to-cell spread.

### 2.4. Comparative Analysis of BoDV-1 Infection in the Culture of the OM and in OECs

To further elucidate the role of OECs during neurotropic viral infections, we compared infection with BoDV-1 in the cultures of rat OM and OECs, by strand-specific qPCR and immunofluorescence. Therefore, the copy numbers of either BoDV-1 gRNA or mRNA were normalized with glycerinaldehyde-3-phosphat-dehydrogenase (GAPDH) mRNA copies. At four dpi, for the OM, a mean of 0.02 normalized BoDV-1-N RNA copies were detectable for both gRNA and mRNA (Figure 6C) In contrast, for the rat OECs, significantly higher copy numbers were obtained. While 0.6 normalized BoDV-1-N RNA copies were measured for the gRNA, a mean of 0.34 copies was found for the respective mRNA. Thus, OECs seem to efficiently replicate BoDV-1 in vitro at early time points after infection. However, at seven dpi, the normalized BoDV-1-N RNA copy numbers decreased for the OECs, and did not significantly differ between the two cell cultures systems. For the OM, 0.14 and 0.26 normalized BoDV-1-N RNA copies were found for the gRNA and mRNA, respectively. Opposing that, 0.25 and 0.21 normalized BoDV-1-N RNA copies were found for the gRNA and mRNA of the OECs, respectively. Although not significant, a slightly higher replication rate seemed to be present in the OECs compared with the cells of the OM. However, the absolute copy numbers of both BoDV-1 RNAs increased from four to seven dpi, but as we found this increase also for GAPDH, which is most likely due to cell proliferation, the normalized copies increased only slightly for the OM and decreased for the OECs (Appendix A).

Moreover, canine and rat OECs were infected with BoDV-1 and examined by immunofluorescence (IF). The rat (Figure 6A) and canine (Figure 6B) OECs were susceptible to infection. The reaction pattern for both cell lines resembled those of the cells of the OM and other cells, as previously described [22]. Distinct granules were seen in the nuclei of a large proportion of the cells. Furthermore, in some cases, a diffuse staining of the cytoplasm was also visible. In two passages, positive cells were counted, and an average of 63% of the infected cells could be determined at six dpi (Appendix A).

## 3. Discussion

Many neurotropic viruses use the olfactory pathway to enter the CNS without the recognition of the immune system [31,32,33]. However, the initial phase of intranasal infection; responsible cell types; and the necessity of BoDV-1 replication, transcription, and translation to ensure successful spread to the brain, remains largely unstudied. Most studies refer to the viral distribution and detection of viral RNA in the CNS [20,21]. The aim of the present study is to characterize the initial phase of an intranasal infection with a neurotropic virus, employing the well-established BoDV-1 rat model. For BoDV-1, the intranasal way of entry via the olfactory system has already been assumed as the most likely natural route of infection. However, intranasal infection with parrot bornavirus was not successful in the experimental studies [34], and it is still not understood how humans get infected, either with BoDV-1 [14] or with VSBV-1 [11]. Detailed knowledge of the early intranasal phase of the infection with neurotropic viruses and responsible cell types is an essential prerequisite in order to be able to develop efficient antiviral prophylactic and therapeutic measures.

### 3.1. Clinic and Histopathology

During our experiments, no clinical signs were observed. This is in accordance with previous studies, where the first clinical signs were found between 18 and 24 days after intranasal BoDV-1 infection [15,21], leading to a mortality rate of up to 90% within one week [21]. However, in a study mimicking natural infection via infectious urine from BoDV-1-infected rats, only ruffled fur, apathy, and bloody noses were reported [15]. It could be speculated that clinical signs might have occurred after a longer duration of infection, but the incubation period may vary, depending on the infectious dose, virus preparation, age, genetic background, and route of administration [35,36,37]. Typically, inflammatory lesions occur shortly before clinical signs [15,37], resulting in an a non-purulent meningoencephalitis [35]. In the present study, only a mild lymphohistiocytic meningitis was present at 21 dpi. However, neither necrotic nor edematous areas within the grey matter were found, as described in a previous study [15]. Interestingly, despite the presence of viral infection, no morphologic changes were detected in the olfactory epithelium, which is consistent with previous studies [15,20,21]. Thus, antiviral cellular strategies, such as degenerative or necrotic lesions, or even inflammation, were not induced.

### 3.2. Characterization of Viral Spread

In this study, BoDV-1-N was detected two days earlier than in a previous study [15]. Interestingly, in the former study, an antigen was detectable in a few of the cells in a time frame between 6 and 20 dpi, and from 22 dpi onwards, no BoDV-1-N was found in the OM. However, the nerve fibers of the lamina propria stayed positive until 27 dpi. In our study, BoDV-1-N was present first in adult ORNs and in their juvenile progenitors, and later, post infection, was detected in all of the cell types and structures of the OM. Moreover, the total number of cells positive for BoDV-1-N increased significantly until 21 dpi. This might indicate the possibility of shedding the virus via nasal secretions during this phase, which might be relevant in possible reservoir species like shrews or squirrels [11,24,25,26]. Whether the viral infection of the OM might have been cleared during the later stages of infection needs to be assessed in future studies.

The presence of viral genomic RNA or mRNA of Mammalian Bornavirus 1 nucleoprotein (BoDV-1-N) in the OM after intranasal infection has not been reported to date. Shankar et al. [20] described the detection of BoDV-1-RNA by RT-PCR. However, the olfactory bulb was only examined as a whole, and RNA was detected at six dpi. In contrast, the BoDV-1 RNA was not detected before 26 dpi in the OM. It cannot be ruled out that the differing findings regarding BoDV-1-N and -RNA were due to the sampling of different areas in the nose or the use of varying infectious doses or virus preparations/passages.

In the present study, viral genomic RNA coding for BoDV-1-N was found later than mRNA coding for BoDV-1-N. This indicates that transcription was turned on before the viral replication started. The detection of mRNA after 14 dpi in cell types other than neurons points to a more delayed viral transcription. It was striking that neither gRNA nor mRNA were found in the sustentacular cells nor in both types of basal cells, despite the presence of BoDV-1-N.

Altogether, the early infection of only a few cells of the OM was sufficient to establish viral infection of the CNS, and viral replication, transcription, and translation took place locally in the OM before the virus reached the brain. The continuous increase in the number of positive neuronal and non-neuronal cells in the OM, often found grouped together, furthermore substantiated the viral spread from cell to cell, as already shown for many other cell types in vitro and in vivo [22,36]. However, the present results argue for a specific role of cells of a neuronal and glial origin for local viral replication, because in both types of ORNs and in the nerve fibers, as well in the OECs, only viral genomic RNA was found. How the virus reaches the brain remains to be elucidated. For a long time it was assumed that retrograde intraaxonal transport represents the major route [38], but for the olfactory neurons, this has to be anterograde transport in the direction of the second olfactory neuron in the main olfactory bulb (MOB) [39]. Moreover, the early and continuous detection of viral products, not only in the nerve fibers, but also in the surrounding OECs, might point to the role of these cells for successful viral entry, possibly by forming channels around the nerve fibers, which could give viral particles access to a free journey to the brain [17,30]. The importance of the OECs for infection via the olfactory route has already been shown for human herpesvirus-6 [35]. In contrast, OECs have been discussed to be an essential component of the innate immune response in order to prevent bacteria from reaching the CNS [40,41,42]. Similar observations have not been made for viral infections to date, and might be different.

BoDV-1-N is the most abundantly expressed viral protein, together with the viral phosphoprotein and an important part of the viral ribonucleoprotein complex (RNP) [43,44,45]. Interestingly, BoDV-1-N was found in all of the cell types of the OM, starting with detection not only in mature, but also in immature ORNs at four dpi. For mature ORNs, infection most likely takes place via the dendrite directly exposed to the nasal cavity. Nevertheless, immature ORNs were also susceptible very early, even though they reached the epithelial surface after their maturation [46]. Thus, they might have been infected by direct contact with the mature ORNs. Viral RNAs were only found in mature and juvenile ORNs, as well as in nerve fibers and OECs, but not in the sustentacular or both types of basal cells. Thus, the presence of only the viral antigen might only reflect the uptake of the viral protein or viral transcription, and replication was below detection levels. For BoDV-1 and the endogenous Bornavirus-like nucleoprotein (EBLN), it has already been shown that an imbalance in the protein amount or an overexpression of components of the RNP prevents superinfection or infection with a closely related virus [47,48]. Thus, this mechanism could play a role in these cells by inhibiting efficient replication. Additionally, the cell type specific differences in the expression of bornaviral endogenous sequences might exist, as there is preliminary evidence that they might be associated with a novel type of antiviral immunity, for example, protection against further Bornavirus infections [49,50,51,52,53].

### 3.3. In Vitro Characterization of the Infection of the OM

The aim of the present study is to establish a cell culture that closely reflects the intranasal infection of adult immunocompetent Lewis rats [15,21]. We refused to use feeder layers, which mostly consist of astrocytes or other glial cells of the Cortex cerebri, and may increase the lifespan of the neurons and help during differentiation [54,55], as glial cells are easily infected with BoDV-1 [22,56,57] and might therefore have biased the results. Thus, we established our culture system, adopting several existing protocols [58,59,60,61]. As reported by others, the number of neurons diminished over time [60], with a complete loss of neurons after seven days in culture, as already reported [58,62]. In our study, the number of cells seeded was defined, and the cells stained with the neuronal marker β III-tubulin were counted and related to the total number of cells in the culture over time. The morphology of the cells and the results from immunoflourescence were comparable to the results of other working groups [63]. A higher proportion of neurons than in the present study was reported by Brauchi et al. [59]. However, a comparison of the results is difficult, because the preparation and culture conditions differed substantially.

### 3.4. BoDV-1-Infection of the Cell Cultures

The BoDV-1 infection of the OM cultures closely resembled the experimental intranasal infection of rats, thereby providing evidence for a reliable in vitro model in order to study many aspects of the early phase of the intranasal infection of neurotropic viruses. This is also of remarkable interest in the context of the 3R principles of Russel and Burch, to contribute to the replacement, reduction, and refinement of animal experiments. However, the susceptibility of cells, especially cell lines, in vitro, does not necessarily reflect the viral tropism in vivo, and therefore findings must be interpreted in the context of viral pathogenesis. A viral antigen was detected at the same time post infection as in the in vivo experiments. IF signals were either intranuclear granules [64] or a diffuse staining of the cytoplasm, as described for the acute infection of astroglial cultures [22,65]. In the present study, a striking aspect was that not only the neurons, but also the non-neuronal components of the OM and the lamina propria were infected, similar to the experimental intranasal infection of rats. Moreover, a spread of the infection was observed over time, while the number of neurons declined. This underscores the fact that not only neurons can be infected.

### 3.5. Comparative BoDV-1-Infection of OM and OECs

To investigate the relevance of the OECs during neurotropic viral infections, as postulated for the human herpesvirus-6 [35], we compared the replication and transcription of BoDV-1 in the cultures of rat OM and OECs by qPCR. At an early stage of infection in vitro, the normalized BoDV-1-N RNA copy numbers were significantly higher in the OECs than in the cells of the OM. Thus, OECs seem to efficiently replicate BoDV-1 in vitro at early time points after infection. However, at later time points, the BoDV-1-N RNA copy numbers decreased for the OECs, and did not differ significantly between the two cell cultures systems. Even though not statistically significant, a slightly higher replication rate seemed to be present in the OECs compared with the cells of the OM, underscoring their potency to efficiently replicate BoDV-1. However, the absolute copy numbers of both BoDV-1 RNAs increased over time, but as we also found this increase for GAPDH, the normalized copies increased only slightly for the OM and decreased for the OECs. This bias regarding GAPDH might in all probability be due to a faster proliferation of OECs, and neuronal loss in the primary culture of the OM over time. The GAPDH housekeeping gene has been shown to work reliably for neuronal cells, and should not be altered by BoDV-1 infection [23,66,67].

Surprisingly, not only the rat but also the canine OECs were susceptible to infection, as determined by IF. The virus suspension used was adapted to the rats, and to date, only a few confirmed infections of a dog with BoDV-1 have been described [68,69]. However, Madin-Darby Canine Kidney (MDCK) cells are also of canine origin, and have been used as persistently infected cells for the diagnostics of the Borna disease (BD) for decades [64]. Strikingly, the canine OECs were highly susceptible, resulting in an average infection rate much higher than the one of the rat OM culture, where the OECs represented only a small share. This underscores their general role during intranasal infection as well as the hypothesis that they promote viral replication as already shown for the human herpesvirus-6 [35]. However, it needs to be mentioned that OECs proliferated under optimal culture conditions, and therefore infection may spread more easily, either by the division of cells or cell-to-cell infection due to shorter distances between cells, as known for cell lines and other primary cultures of the CNS [22,65,70].

In summary, the present study provides in vivo and in vitro evidence for the important role of OM for the initial viral transcription, replication, and translation with olfactory neurons and OECs as potential key players. The initial infection of only a few cells is sufficient for enabling a successful spread to the brain, and no morphological alterations are induced, so BoDV-1 has developed a perfect strategy to infect new hosts by the olfactory route, circumventing the host’s immune system. However, many open questions regarding the immune response of the olfactory system and its impact on species specificity remain to be solved.

## 4. Materials and Methods

### 4.1. Virus Stock Preparation

The procedures for the in vivo experiments were approved by the Committee on the Ethics of Animal Experiments of the Regional Council of Giessen (permit no. V54-19 c 2015(1) GI 18/4 Nr. 102/2011, approval date 24 February 2012). Brain homogenate (BV-E7 NB-4P/2, 24.03.1988; strain He/80; titer 6 × 10^6^ ID_50_/mL; generous gift of Dr. S. Herzog, Institute of Virology, Justus-Liebig University, Giessen [19]) was used as the virus stock and was passaged once in newborn Lewis rats. Therefore, the rats were infected intracranially with a dose of 6000 ID_50_, and the brain tissue was harvested and homogenized by ultrasound together with Dulbecco’s Modified Eagle Medium (DMEM) (low glucose, Gibco, Carlsbad, CA, USA), containing 2% fetal calf serum (FCS, PAA, Pasching, Austria). After centrifugation, the supernatant was stored at −80 °C.

### 4.2. Intranasal Infection of Rats

Four week-old Lewis rats were kept in individually ventilated cages at the BSL-3 animal facility at the Institute of Virology, Philipps University Marburg. The rats were infected intranasally with a dose of 40,000 ID_50_ diluted in 200 µl DMEM, as previously described [19]. The control groups included two mock-infected rats, only receiving DMEM. They were monitored daily for clinical signs and underwent extensive medical examination every three to four days until 14 dpi, and every other day for time points beyond 14 dpi. The rats were euthanized by cervical dislocation under deep isoflurane anesthesia at 3 and 18 hpi, as well as at 1, 2, 4, 7, 14, and 21 dpi. At each time point, five rats, and at 21 dpi, two rats, were examined.

### 4.3. Tissue Preparation and Histology

The heads were skinned, fixed for one week in 10% non-buffered formalin, and cut with a diamond band saw, with the first coronal cut at the transition between the olfactory bulb and ethmoid, resulting in seven to eight slices, depending on the age and size of the rat. The tissue was decalcified in ethylenediaminetetraacetic acid (EDTA) at 37 °C for seven days, and incubated in tap water over night prior to routine paraffin embedding. Sections of 4 µm thickness were hematoxylin and eosin (H&E) stained in an autostainer (Microm, HMS 740). The slides were scanned by light microscopy for inflammatory or degenerative changes related to BoDV-1 infection.

### 4.4. Immunohistochemistry

The monoclonal Bo18 antibody directed against the BoDV-1 nucleoprotein was a generous gift from Dr. Sibylle Herzog, Institute of Virology, Faculty of Veterinary Medicine, Justus-Liebig University Giessen. The brain sections of a naturally BoDV-1-infected horse were taken as the positive control. The mouse monoclonal antibody “T1” against chicken lymphocytes served as the negative control [71]. The slides were deparaffinized in xylol and a descending alcohol series. Endogenous peroxidase was blocked by incubation in methanol, with 0.05% H_2_O_2_ for 30 min. To avoid non-specific binding, the slides were incubated for 30 min with 20% nonimmune horse serum (PAA Laboratories GmbH) diluted in Tris-buffered saline (TBS), supplemented with 1% bovine serum albumin (BSA). The primary antibody and negative control antibody were diluted 1:500 in TBS containing 1% BSA, and were incubated on the slides over night at 4 °C in a humid chamber. A biotinylated horse anti-mouse antibody (cat no. BA-2000, Vector Laboratories, Inc., Burlingame, CA, USA) was used as the secondary antibody (9 µL/1000 µL TBS + 1% BSA) and the Vectastain^®^ ABC Kit Peroxidase Standard (cat no. PK-4000, Vector Laboratories, Inc., Burlingame, CA, USA) served as the detection system. After several washing steps, the slides were incubated in DAB (3,3-diaminobenzidine-tetrahydrochloride, Sigma-Aldrich, St. Louis, MO, USA) and H_2_O_2_ in 0.1M imidazole, pH 7.1 for 3 min, to visualize the specific antigen–antibody binding and counterstained with Papanicolaou stain.

### 4.5. In Situ Hybridization (ISH)

Two digoxigenin-coupled RNA probes were constructed either against the viral genome (sense—negative strand), coding for BoDV-1-N, or against the corresponding mRNA (antisense—positive strand) based on a protocol previously published [72,73]. Briefly, the plasmid containing BoDV-1-N was obtained by the use of the TOPO TA Cloning^®^ Kit for Sequencing (Life Technologies, Carlsbad, CA, USA) containing the vector pCR™4-TOPO^®^ TA and One Shot^®^ TOP10 Chemically Competent *E. coli*. The insert originates from the natural strain H24, and reveals a length of 342 bp (GenBank^®^ accession number AF158629, Position: 87-428). The plasmid DNA was isolated with nexttec^TM^ cleanColumns (nexttec^TM^ DNA isolation systems, Hilgertshausen, Germany), according to the manufacturer’s instructions, and was amplified. In order to detect the genomic RNA, the BoDV-1-N antisense primer (5’-AAT GAG CAA CAA TGG CTG AA-3’) was used together with an M13 forward primer. To obtain a probe that recognizes the mRNA, the BoDV-1-N sense primer (5’-CCC CGG AAA ATT CCT ACA AT-3’) was used in combination with an M13 reverse primer. Cycling was performed on a Multicycler^®^ PTC 200 (Biozym Diagnostik GmbH, Hessisch Oldendorf, Germany) with the following steps: initial denaturation at 94 °C for 2 min, 34 cycles consisting of denaturation at 94 °C for 30 s, annealing for 30 s at 50 °C, and elongation for 1 min at 72 °C. Afterwards, a final elongation for 10 min at 72 °C followed. The PCR products were cleaned up to remove the spare primers and nucleotides with the Amicon Ultra Centrifugal Filters 0.5 mL (Merck Millipore, Darmstadt, Germany), according to the manufacturer’s instructions. The DNA amount was determined with the Nanodrop^®^ 2000 (Thermo Fisher Scientific Inc., Waltham, MA, USA). In vitro transcription was carried out with the DIG RNA Labeling Kit (SP6/T7) and T3 RNA polymerase (both Roche Diagnostics, Mannheim, Germany), according to the manufacturer’s instructions.

ISH was performed according to previously published protocols [36,72,73]. Briefly, 4-µm thick tissue sections on glass slides (Superfrost Plus^®^, R. Langenbrinck, Emmendingen, Germany) were deparaffinized and rehydrated in a descending alcohol series. Afterwards, proteolytic digestion, post-fixation, acetylation, and prehybridization followed. Hybridization of the probes was carried out overnight in a humid chamber at 52 °C. After several washing steps, non-bound RNA was removed by incubation with a mixture of RNases A and T (Roche diagnostics). As a detection system, an anti-DIG-antibody coupled with alkaline phosphatase (Roche diagnostics) was used in combination with the substrates nitroblue tetrazoliumchloride (NBT) and 5-bromo-4-chloro-3-indolyl phosphate (BCIP; both from Sigma-Aldrich). Finally, the slides were mounted with Kaisers Glyceringelatine (Merck, Darmstadt, Germany). Each sample was incubated, with the probe detecting genomic or mRNA, and the hybridization mix only to exclude unspecific signals. As the positive control, the brain slides of a naturally infected horses were treated similarly.

### 4.6. Evaluation and Statistical Analysis of Viral Antigen and RNA Detection by Immunohistochemistry and In Situ Hybridization

The tissues of the infected animals were evaluated by light microscopy with a 20× objective, according to the following scheme for the semiquantitative description of the number of positive cells in total: score 0—no stained cells or structures on the slide; score 1—up to 10 stained cells on the slide; score 2—between 10 and 60 stained cells scattered on the slide or located in nests; and score 3—more than 60 stained cells scattered on the slide or located in nests. This scheme was applied for the demonstration of viral antigen and RNA by immunohistochemistry and in situ hybridization, respectively. In addition, the positive cells of the OM and the nerve fibers in the lamina propria were evaluated separately as follows: adult and juvenile ORNs, sustentacular cells, globose and horizontal basal cells, nerve fibers, and olfactory ensheathing cells in the lamina propria using the score described above and by typical morphological appearance. A positive immunohistological staining was characterized by single or multiple intranuclear brown granules, or a patchy to diffuse brown staining of the cytoplasm, including axon and dendrite. The specific signal depicted by ISH consisted of few distinct dark blue to purple granules in the nucleus, or a granular to diffuse staining of the cytoplasm of the same color.

To determine whether the detection of the BoDV-1-N or the corresponding mRNA and gRNA changed significantly in a time-dependent manner, an exact Kruskal–Wallis test was performed with StatXact (Cytel Studio, version 9.0.02010, Cambridge, MA, USA). Furthermore, to examine whether the differences in BoDV-1-N, mRNA, and gRNA detection followed a monotonic trend, the Spearman’s rank correlation was carried out using BMDP (Dixon, Release 8.1, 1993, Los Angeles, CA, USA). All of the analyses were performed by the working group Biomathematik und Datenverarbeitung, of the Faculty of Veterinary Medicine of the Justus Liebig University of Giessen. All of the results were accepted as statistically significant for *p* ≤ 0.05.

### 4.7. Culture of the Rat Olfactory Mucosa

To examine if the cells of the OM can be infected with BoDV-1 in vitro, a culture of the rat OM was established. The rats were anesthetized with CO_2_ and were decapitated. The heads were skinned, and the nasal turbinates and caudal third of the nasal septum were enzymatically digested. Thereafter, the tissue was placed in DMEM (low glucose, without any supplementation, Gibco, Carlsbad, CA, USA) with 1 mg/mL collagenase, 2.4 U/mL dispase II, and 50 µg/mL DNase II (all Sigma-Aldrich), and incubated in a heat chamber at 37 °C on a shaking platform at 80 rpm for 1 h. Afterwards, the tissue was carefully washed with prewarmed DMEM without any supplement, and triturated with Neurobasal^®^-A medium containing a B-27^®^ supplement (both Gibco) and a fire-polished glass pasteur pipette. The supernatant was removed to allowed the cells to sink for 5 min. The remaining supernatant was removed and replaced by a fresh Neurobasal^®^-A medium with a B-27^®^ supplement for seeding on glass coverslips (20 × 20 mm; Carl Roth, Karlsruhe, Germany), coated with poly-L-lysine, according to the protocol by Ahlemeyer et al. [74]. About 250,000 cells per coverslip were seeded. For the initial plating of the cells, GlutaMAX^TM^ Supplement (Gibco^®^) was added, and after the adhesion of the cells, the culture medium was changed to a Neurobasal^®^-A medium with a B-27^®^ supplement. The cultures were kept at 37 °C in a humid chamber at 5% CO_2_, and the culture medium was changed every other day. All of the culture media were supplemented with 1% penicillin/streptomycin (Gibco^®^) and 0.1% gentamicin (PAA Laboratories, Pasching, Austria).

### 4.8. Cell Culture of Purified Adult Canine and Rat Olfactory Ensheathing Cells (OECs)

Schwann cell-free canine OECs derived from the olfactory mucosa of adult dogs were kept under conditions as previously published [75,76]. The rat OECs (Rolf B1.T (ECACC 03071601)) [77] were obtained from Merck KGaA (Darmstadt, Germany), and handled according to the manufacturer’s instructions.

### 4.9. BoDV-1-Infection of the Cell Cultures

The cell culture of the rat olfactory mucosa and the cultures of the canine and rat olfactory ensheathing cells were infected for 1 h at 37 °C with a multiplicity of infection (MOI) of 0.8 to 1. Afterwards, the cultures were washed for 5 min with prewarmed medium, before 2 mL of fresh culture medium was added.

### 4.10. Immunofluorescence (IF) of the Cell Cultures

IF was performed in order to estimate the amount of mature olfactory neurons in the culture of the OM, and to confirm BoDV-1 infection. The ells were fixed with 4% paraformaldehyde (PFA) in a culture medium and permeabilized with 5% nonimmune goat serum (NGS), 3% BSA, and 0.25% (for TUJ1 staining) or 0.1% (for Bo18 staining) Triton^®^ X-100 for 20 min. For the staining of the olfactory neurons, anti- β III tubulin (TUJ1; Covance, Dedham, MA, USA) was used at a dilution 1:1000, in Tris buffered saline (TBS) containing 3% bovine serum albumin (BSA) and 5% nonimmune goat serum (NGS). To detect BoDV-1-N, the same monoclonal antibody against BoDV-1-N (Bo18) was used as described for immunohistochemistry, diluted 1:100 in TBS containing 1% BSA. The cells were incubated with secondary antibodies (Cy2 goat anti-rabbit and Cy3 goat anti-mouse, respectively; Jackson ImmunoResearch, West Grove, PA, USA) for 30 min. Then, 4’,6-diamidino-2-phenylindole (DAPI; Carl Roth, Karlsruhe, Germany) was used to stain the nuclei. Images were taken with Nikon Eclipse 80i fluorescence microscope and NIS Elements BR 3.2 software. The coverslips were mounted with Entellan in Toluen (Merck, Darmstadt, Germany).

### 4.11. Characterization of the Culture of the OM

As the main ORNs were of particular interest for the study, their percentage was determined by counting the β III-tubulin-positive cells related to the total number of cells in the culture. The remaining cells were determined by their morphology. For this, the cells were fixed for 4 and 24 h, as well as two, four, and seven days after plating with 4% PFA in a culture medium. The cells were stained with β III-tubulin, according to the protocol described above. Per timepoint, three coverslips with at least 200 cells each were counted. The infected cultures were fixed 4, 7, 10, and 14 dpi, and the IF was done as previously described. Per timepoint, at least 600 cells were counted and checked for infection. All of the experiments were carried out three times.

### 4.12. Quantitative Real Time RT-PCR to Determine Viral Replication (BoDV-1 gRNA) and Transcription (BoDV-1 mRNA)

To elucidate the role of OECs on the replication and transcription of neurotropic viruses, we used BoDV-1 as a model and infected the primary culture of rat OM and OECs. Therefore, the culture of rat OM was slightly modified by centrifugation for 3 min at 125 g. To avoid an overgrowth of proliferating OECs, the cell cultures were slowly adapted to a serum-free culture medium. Quantitative real time RT-PCR (qPCR) to determine the viral replication and transcription was used according to previously published protocols. Therefore, viral genomic RNA (gRNA) mRNA was quantified and normalized with rat GAPDH as the housekeeping gene.

The RNA was isolated four and seven days after the infection, with a protocol based on Chomczynski and Sacchi [78]. To purify the RNA and to remove the proteins and small RNA fragments, the RNeasy Mini Kit (Qiagen, Venlo, The Netherlands) was used according to the manufacturer’s instructions, and DNase I digestion was also performed. Finally, the RNA was resuspended in 30 µl of RNase-free water and quantified with a Nanodrop 2000 spectrophotometer.

To obtain cDNA, the RT reaction was done with the QuantiTect^®^ Reverse Transcription Kit (Qiagen) and the Multicycler^®^ PTC 200. Only specific primers were used in this reaction, and an additional step for the annealing of the specific primer was inserted. This was done after the wipeout step of the genomic DNA for 5 min at 70 °C. For the detection of the genomic viral RNA and the mRNA, the corresponding sense and antisense primer, respectively, were used. For the detection of the corresponding mRNA from the housekeeping gene, GAPDH, the antisense primer was used (Appendix A).

For qPCR, 5’-FAM-3’-TAMRA- and 5’-Hex-3’-BHQ-1-labeled TaqMan^®^ probes (Eurogentec SA, Liège, Belgium) were used, and PCR was carried out in the Rotor-Gene^®^ Q thermocycler with the Rotor-Gene^®^ Probe PCR Kit (both Qiagen). The copy numbers were calculated in relation to the standard dilution series for 10^3^ to 10^8^ copies for BoDV-1-N and 10^2^ to 10^8^ copies for GAPDH, as previously published [23,79]. For each reaction, 11 µL of the cDNA was inserted in the PCR. Uninfected samples served as a negative control, and no template controls were used in each run. After the activation of the HotStarTaq Plus^®^ DNA Polymerase for 3 min at 95 °C, the 40 cycling steps were as follows: denaturation at 95 °C for 3 s and annealing/elongation at 60 °C for 10 s.

Each sample was normalized by calculating the quotient of the geometric mean BoDV-1-specific copy number and the number of the copies GAPDH from duplicates of three coverslips. The significance of the normalized copies was calculated using an unpaired t-test with GraphPad Prism Version 7 (GraphPad software, San Diego, CA, USA).

## Figures and Tables

**Figure 1 ijms-20-01318-f001:**
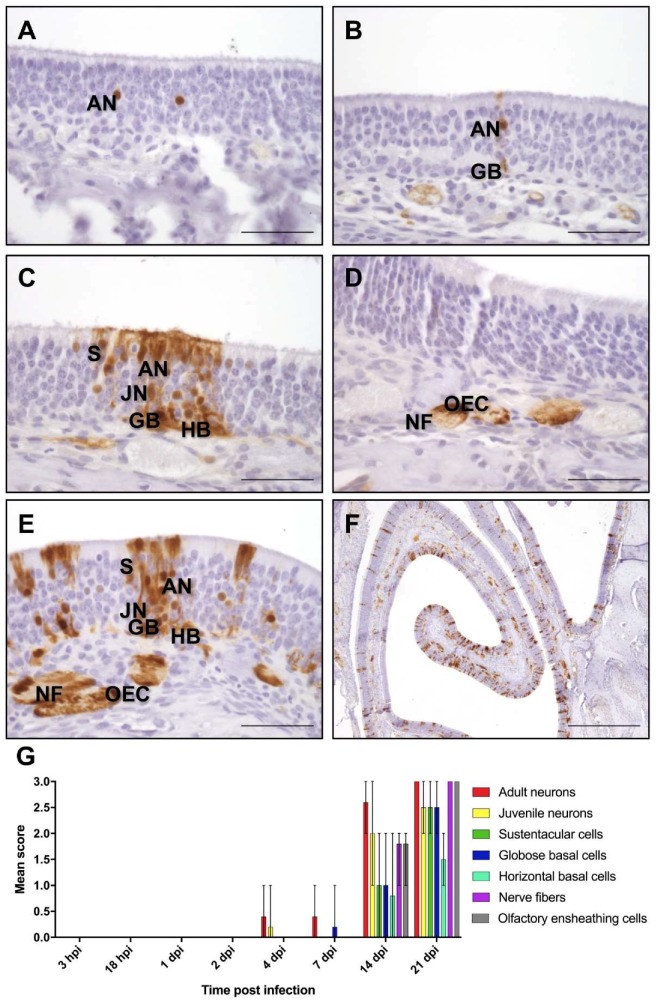
Immunohistological detection of Mammalian Bornavirus nucleoprotein (BoDV-1-N) in the olfactory mucosa (OM). At (**A**) 4 days post infection (dpi); (**B**) 7 dpi; (**C**,**D**) 14 dpi; and (**E**,**F**) 21 dpi. AN—adult neurons; JN—juvenile neurons; GB—globose basal cells; HB—horizontal basal cells; NF—nerve fibers; OEC—olfactory ensheathing cells. Scale bar (**A**–**E**): 50 µm; (**F**) 500 µm. (**G**) Mean score of immunohistochemistry: arithmetic mean; error bars: min/max; hpi—hours post infection.

**Figure 2 ijms-20-01318-f002:**
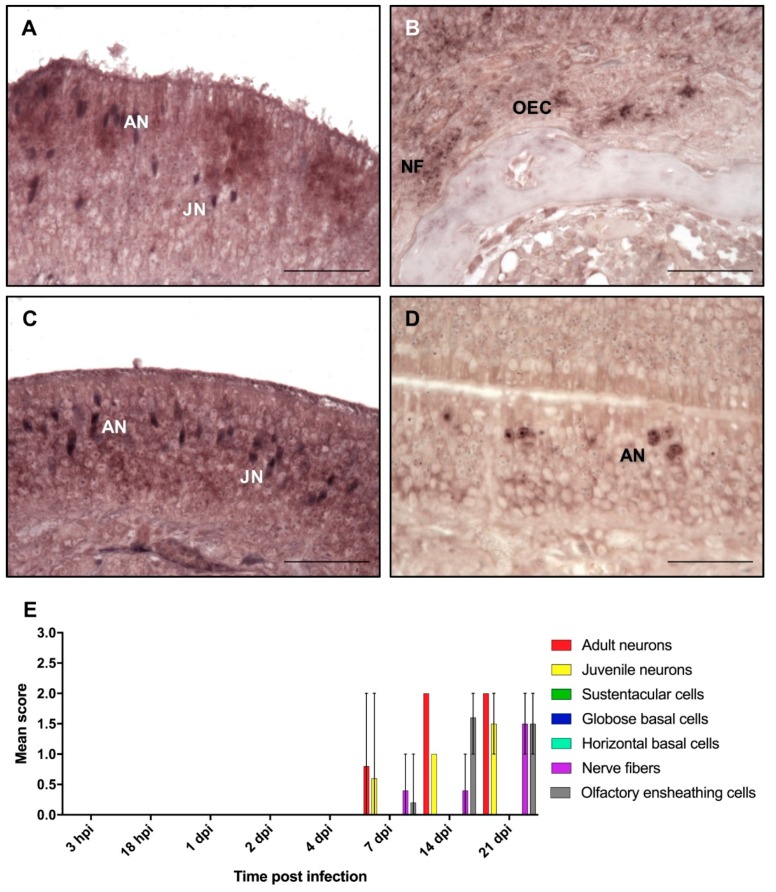
In situ hybridization for the detection of viral genomic RNA in the OM. At (**A**) 7 dpi; (**B**) OECs and nerve fibers located in the lamina propria at 7 dpi; (**C**) 14 dpi; (**D**) 21 dpi. AN—adult neurons; JN—juvenile neurons; NF—nerve fibers; OEC—olfactory ensheathing cells. Scale bar (**A**–**D**): 50 µm. (**E**) Mean score of in situ hybridization: arithmetic mean; error bars: min/max.

**Figure 3 ijms-20-01318-f003:**
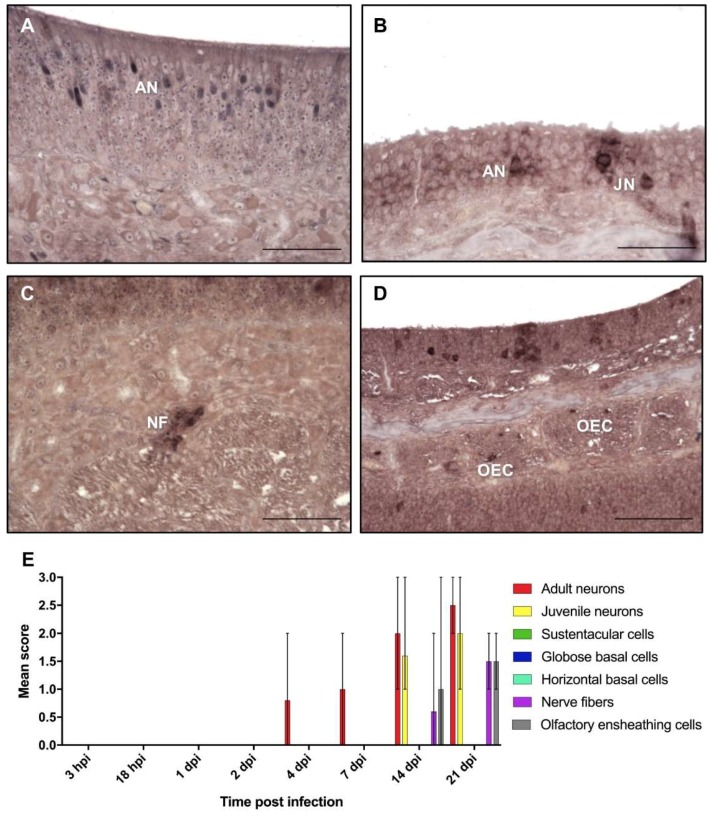
In situ hybridization for the detection of messenger RNA in the olfactory mucosa. At (**A**) 4 dpi; (**B**) 14 dpi; (**C**) nerve fibers located in the lamina propria at 21 dpi; (**D**) 21 dpi. AN—adult neurons; JN—juvenile neurons; NF—nerve fibers; OEC—olfactory ensheathing cells. Scale bar (**A**–**D**): 50 µm. (**E**) Mean score of in situ hybridization: arithmetic mean; error bars: min/max.

**Figure 4 ijms-20-01318-f004:**
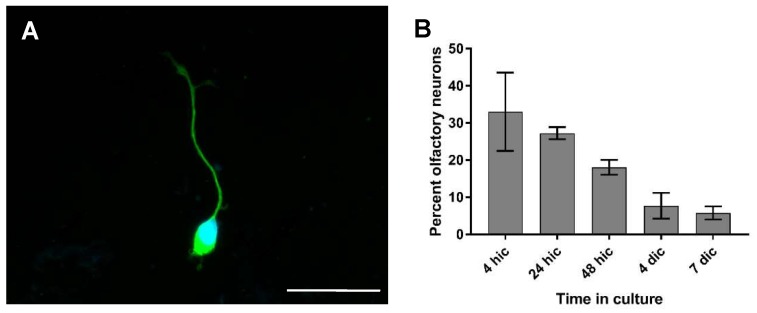
Immunofluorescence of the OM culture for the detection of olfactory receptor neurons (ORNs). (**A**) Olfactory neuron after two days in culture (ic). β III-Tubulin—Cy2; nuclear staining: 4’,6-diamidino-2-phenylindole (DAPI); ic—in culture. Scale bar: 20 µm. (**B**) Percentage of olfactory neurons in the primary culture of the OM over time. Arithmetic mean, error bars: standard deviation.

**Figure 5 ijms-20-01318-f005:**
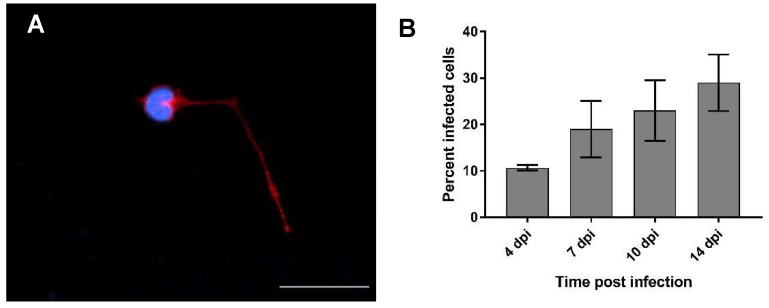
Percentage of BoDV-1-infected cells in the OM culture over time. (**A**) Olfactory neuron at seven dpi. Bo18—Cy3. Nuclear staining: DAPI. Scale bar: 20 µm. (**B**) Percentage of infected cells in the OM culture over time. Arithmetic mean, error bars: standard deviation.

**Figure 6 ijms-20-01318-f006:**
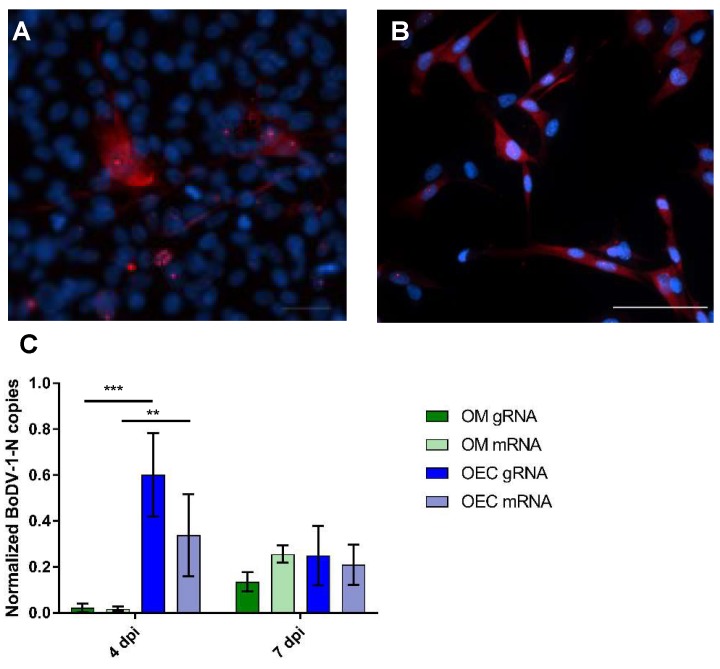
Infection of the OM culture and OECs. (**A**) Rat OECs at five dpi; (**B**) Canine OECs at six dpi. (**A**,**B**) Immunofluorescence—Bo18: Cy3. Nuclear and cytoplasmatic staining: DAPI. Arrows: BoDV-1-N. Insert: magnification of a cell at six dpi. Scale bars: 50 µm. Arrows: intranuclear granular staining for BoDV-1. (**C**) Quantitative real time RT-PCR, normalized Mammalian Bornavirus 1 nucleoprotein BoDV-1-N copy numbers at four and seven dpi for viral genomic RNA (gRNA) and messenger RNA (mRNA) for the cultures of the olfactory mucosa (OM) and for rat olfactory ensheathing cells (OECs), respectively. Arithmetic mean, error bars: standard deviation. ** *p* = 0.0080, *** *p* = 0.0002.

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
