# Peer review of "Intranasal Borna Disease Virus (BoDV-1) Infection: Insights into Initial Steps and Potential Contagiosity"

_ijms, 2019, doi:10.3390/ijms20061318_

Round 1
Reviewer 1 Report
In this manuscript, Kupke et al provide a precise description of the modalities of BoDV intranasal infection. This paper brings important novel information about this mode of contamination. Considering the recent demonstration of the zoonotic potential of Orthobornaviruses, this study seems particularly timely and appropriate, especially because very little in known about the modalities of BoDV infection by the intranasal route, which is, however, the most likely frequent route of contamination. A better understanding of the initial stages of infection may provide novel information on the first cellular targets, preceding neuronal spread and resulting encephalitis and, as such, may provide clues for therapeutic intervention.
I am therefore convinced of the interest of this work. I also found that this study was extremely carefully and precisely done and the results are solid. I, however, was less thrilled by the "packaging" of the story. Indeed, the manuscript is often difficult to follow and could gain a lot by being simplified and shortened. Because the authors try to be very precise and exhaustive, I think that they end up by "drowning" the reader in an "avalanche" of details that diverge from the initial message and eventually "dilute" the mains conclusions of the work. The English spelling would also benefit from substantial simplification (by chasing and removing all unnecessary words in their sentences). I am convinced that such a simplification effort would make their story punchier and more efficient.
Specific points:
1. I could not find any mention about the viral strain used, its origin, and how virus stocks were prepared and diluted. In my experience, tissue culture-adapted viruses do not display the same properties when next inoculated in vivo. It is preferable to use a viral strain that was only passaged in animal brains. In the latter case, however, the preferred control should consist of a purified brain homogenate from a non-infected animal. The reference cited in the methods about this point (19) does not seem to be related to this group.
2. There are large sections of the introduction that are actually presented as a discussion. It would be better to condense the introduction part, to just introduce the model and the question addressed. Accordingly, the discussion section is again too long, sometimes repeating results. These two parts of the manuscript would really need to be revised for succinctness and clarity.
3. I think that the data about immunodetection of BoDV-N should be presented differently. Indeed, the authors should refrain from presenting their results as "epiphenomena" (one positive staining being detected in 2/5 rats….). I would suggest that the authors present a table for the different animals tested and show the score (or positivity) for each of them over time. What the readers need to see is the global pattern of BoDV staining upon IN infection.
4. The authors should better justify why they decided to assess BoDV presence using both IHC and ISH (sensitivity? Other reason?). Also, I did not catch what was the need to perform correlative statistics about IHC and ISH data.
5. Regarding the in vitro OM cultures, these are "dissociated" not "dissociative". The data are interesting. However, the authors should comment on the fact that BoDV infection in vitro does not necessarily reflect the in vivo situation. For instance, Vero cells are often used, although they do not reflect the original viral tropism.
Author Response
In this manuscript, Kupke et al provide a precise description of the modalities of BoDV intranasal infection. This paper brings important novel information about this mode of contamination. Considering the recent demonstration of the zoonotic potential of Orthobornaviruses, this study seems particularly timely and appropriate, especially because very little in known about the modalities of BoDV infection by the intranasal route, which is, however, the most likely frequent route of contamination. A better understanding of the initial stages of infection may provide novel information on the first cellular targets, preceding neuronal spread and resulting encephalitis and, as such, may provide clues for therapeutic intervention.
I am therefore convinced of the interest of this work. I also found that this study was extremely carefully and precisely done and the results are solid. I, however, was less thrilled by the "packaging" of the story. Indeed, the manuscript is often difficult to follow and could gain a lot by being simplified and shortened. Because the authors try to be very precise and exhaustive, I think that they end up by "drowning" the reader in an "avalanche" of details that diverge from the initial message and eventually "dilute" the mains conclusions of the work. The English spelling would also benefit from substantial simplification (by chasing and removing all unnecessary words in their sentences). I am convinced that such a simplification effort would make their story punchier and more efficient.
We thank the reviewer for this useful comment. We will shorten and simplify the manuscript according to the reviewer’s suggestion.
Specific points:
1. I could not find any mention about the viral strain used, its origin, and how virus stocks were prepared and diluted. In my experience, tissue culture-adapted viruses do not display the same properties when next inoculated in vivo. It is preferable to use a viral strain that was only passaged in animal brains. In the latter case, however, the preferred control should consist of a purified brain homogenate from a non-infected animal. The reference cited in the methods about this point (19) does not seem to be related to this group.
The authors would like to thank the reviewer for this good indication. Indeed, brain homogenates derived from either neonatally infected or non-infected rats was used. We will clarify this in the materials and methods section. The working group we cited is closely related to our working group but since most members are not working actively anymore this relation might not be very obvious.
2. There are large sections of the introduction that are actually presented as a discussion. It would be better to condense the introduction part, to just introduce the model and the question addressed. Accordingly, the discussion section is again too long, sometimes repeating results. These two parts of the manuscript would really need to be revised for succinctness and clarity.
The authors agree with the reviewer in both points and will shorten both sections as much as possible.
3. I think that the data about immunodetection of BoDV-N should be presented differently. Indeed, the authors should refrain from presenting their results as "epiphenomena" (one positive staining being detected in 2/5 rats….). I would suggest that the authors present a table for the different animals tested and show the score (or positivity) for each of them over time. What the readers need to see is the global pattern of BoDV staining upon IN infection.
We thank the reviewer for the suggestion to present the data as a table and we would like to include this table in the supplement section. This will allow us to shorten the description of the findings. However, the table alone might not be very lucid since we need to include all the timepoints, cell types and animals. Therefore, we would like to keep the diagrams to keep the overview.
Nevertheless, we believe that it is necessary to highlight the positive staining of few cells at the very beginning in few animals since these findings are contradictory to the results of other working groups. It is an important finding that for BoDV-1 replication, transcription and translation takes place in the olfactory mucosa at very early timepoints after infection before a spread to CNS occurs. Why other studies have failed to detect BoDV-1 in the olfactory mucosa is not fully clear but we believe that it is due to methodology or sensitivity.
4. The authors should better justify why they decided to assess BoDV presence using both IHC and ISH (sensitivity? Other reason?). Also, I did not catch what was the need to perform correlative statistics about IHC and ISH data.
The authors are grateful for this constructive comment. We have chosen both methods to examine and compare where replication, transcription (detection of RNA, ISH) and translation (detection of viral antigen IHC) takes place. It could have been possible that the olfactory mucosa and the olfactory nerve just function as a transport vehicle to the CNS without noteworthy viral multiplication like it is described for e.g. rabies virus. If that would have been the case, we would not have been able to detect an extensive staining for both ISH probes. Therefore, we have chosen both methods.
The correlative statistics (Spearman’s rank correlation) was additionally performed to confirm statistically that the relationship between timepoint and number of positive cells followed a monotonic trend. That means that positive cells of the distinct cell types increase over time and do not stagnate or decrease until the end of the study.
We will address both remarks in the manuscript.
5. Regarding the in vitro OM cultures, these are "dissociated" not "dissociative". The data are interesting. However, the authors should comment on the fact that BoDV infection in vitro does not necessarily reflect the in vivo situation. For instance, Vero cells are often used, although they do not reflect the original viral tropism.
We fully agree with the reviewer. We will change the terminology in the manuscript and include comments on the in vitro/in vivo situation in the discussion.
Reviewer 2 Report
The paper on Borna disease virus infection is very interesting, mainly due to the fact, that it shows and proves new pivotal elements in the transmission of this disease that may serve as a model or an example in other neutrotropic viral infections with zoonotic potential. I believe that the crucial part of this study is showing the role of olfactory ensheating cells for infection via the olfactory pathway, which has never been analyzed for any other virus than HHV-6 and that is why I would like to read a sentence or two more about the potential role of those cells in the Introduction.
The Introduction is well written, good for specialists and understandable for those who are not in depth of this problem. I think the experiment is well planned and the methods are perfectly chosen. The results are convincing and the summarizing of the Authors is very mature.
I recommend this paper to be printed.
Author Response
The paper on Borna disease virus infection is very interesting, mainly due to the fact, that it shows and proves new pivotal elements in the transmission of this disease that may serve as a model or an example in other neutrotropic viral infections with zoonotic potential. I believe that the crucial part of this study is showing the role of olfactory ensheating cells for infection via the olfactory pathway, which has never been analyzed for any other virus than HHV-6 and that is why I would like to read a sentence or two more about the potential role of those cells in the Introduction.
The Introduction is well written, good for specialists and understandable for those who are not in depth of this problem. I think the experiment is well planned and the methods are perfectly chosen. The results are convincing and the summarizing of the Authors is very mature.
I recommend this paper to be printed.
The authors would like to thank the reviewer for the kind comments and will include the role of the olfactory ensheathing cells in the introduction.